# Early postpartum HbA1c after hyperglycemia first detected in pregnancy—Imperfect but not without value

**Ankia Coetzee**[1]*, **David R. Hall**[2], **Mari van de Vyver**[3], **Magda Conradie**[2]

**1** Department of Medicine, Division of Endocrinology Stellenbosch University and Tygerberg Hospital, Cape Town, South Africa, **2** Department of Obstetrics & Gynecology, Stellenbosch University and Tygerberg Hospital, Cape Town, South Africa, **3** Department of Medicine, Division of Clinical Pharmacology, Stellenbosch University and Tygerberg Hospital, Cape Town, South Africa

* ankiac@sun.ac.za

## Abstract

### Background

South African **w**omen of childbearing age are disproportionally affected by obesity and at significant risk of Type 2 Diabetes Mellitus (T2DM). Unless pregnant, they do not readily undergo screening for T2DM. With a local focus on improved antenatal care, hyperglycemia is often first detected in pregnancy (HFDP). This may erroneously be attributed to Gestational Diabetes Mellitus (GDM) in all without considering T2DM. Glucose evaluation following pregnancy is essential for early detection and management of women with T2DM in whom persistent hyperglycemia is to be expected. Conventional testing with an oral glucose tolerance test (OGTT) is cumbersome, prompting investigation for alternate solutions.

### Aim

To compare the diagnostic performance of HbA1c to the current gold standard OGTT in women with HFDP 4–12 weeks post-delivery.

### Methods

Glucose homeostasis was assessed with OGTT and HbA1c in 167 women with HFDP, 4–12 weeks after delivery. Glucose status was based on American Diabetes Association criteria.

### Results

Glucose homeostasis was assessed at 10 weeks (IQR 7–12) after delivery. Of the 167 participants, 52 (31%) had hyperglycemia, which was comprised of 34 (20%) prediabetes and 18 (11%) T2DM. Twelve women in the prediabetes subgroup had diagnostic fasting plasma glucose (FPG) and 2-hour plasma glucose (2hPG), but in two-thirds of the patients (22/34) only one time point proved diagnostic. The FPGs and the 2hPGs of six women with HbA1c-based T2DM were both within the prediabetes diagnostic range. According to the HbA1c measurements, 85% of 52 participants with gold standard OGTT defined hyperglycemia

**Data Availability Statement:** All relevant data are within the manuscript and its Supporting information files. Data have also been deposited at

the following link: (https://figshare.com/s/
63198b5802c6fa17fac4).

**Funding:** The author(s) received no specific
funding for this work.

**Competing interests:** The authors have declared
that no competing interests exist.

(prediabetes and T2DM) as well as 15 of 18 women with postpartum persistent T2DM were correctly classified. According to FPG, 15 women with persistent hyperglycemia would have been missed (11 with prediabetes and four with T2DM; 29%). When compared to an OGTT, a single HbA1c of 6.5% (48mmol/mol) postpartum demonstrated a sensitivity of 83% and specificity of 97% for the identification of T2DM.

## Conclusion

HbA1c may improve access to postpartum testing in overburdened clinical settings where the required standards of OGTT cannot be guaranteed. HbA1c is a valuable test to detect women who will benefit most from early intervention but cannot unequivocally replace OGTT.

## Introduction

Increasing prevalence of Type 2 Diabetes Mellitus (T2DM) in reproductive-aged women, in part driven by obesity, has contributed to the current high rate (1 in 6) of births with hyperglycemia [1–4]. In low-to-middle income countries (LMIC), such as South Africa, women of childbearing age are disproportionally affected by obesity, predisposing them to Gestational Diabetes Mellitus (GDM) and T2DM [5–8]. T2DM screening is limited in the local community outside of pregnancy, with two-thirds of those identified being unaware of their diagnosis [9–12]. To improve antenatal care, most international and some South African guidelines advocate universal GDM screening. Due to resource limitations, selective screening of women with the highest risk for T2DM remains the norm locally [12–15]. Hyperglycemia first detected in pregnancy (HFDP) which meets the criteria for overt T2DM, is often mistaken for GDM [15–17]. T2DM is expected to persist post-delivery, whereas GDM characteristically resolves (within 24–48 hours) postpartum [18–20]. Women, with GDM, however, remain at very high risk of future T2DM [18, 19].

The early detection and optimal management of T2DM has long-term benefit and is referred to as the so called "legacy effect" [21–23] Landmark studies support the hypothesis that progression to T2DM is preventable in women with GDM [24, 25]. In the Diabetes Prevention Program Outcomes Study, intensive lifestyle modifications and metformin compared to placebo, in women with prior GDM, reduced the incidence of T2DM by 35% and 40%, respectively. Against the backdrop of the metabolic "legacy effect" several professional organizations, including the World Health Organisation (WHO) and the International Association for Diabetes in Pregnancy Study Groups (IADPSG), have earmarked the early postpartum period as a window of opportunity to firstly assess glucose status after HFDP and then to ensure early intervention if indicated [26–28]. Early postpartum glucose evaluation is standard practice at our institution, where as many as 26% of women remain hyperglycemic following delivery [17].

The 75-gram oral glucose tolerance test (OGTT) is the gold standard for postpartum glucose evaluation. It is a cumbersome test that requires fasting, glucose ingestion and a minimum of 2-hours at a healthcare facility [29]. Glycated hemoglobin (HbA1c), the monitoring modality for longer-term glycemia, is an alternate and convenient diagnostic tool to identify non-pregnant people with T2DM and prediabetes [12, 29]. HbA1c testing requires a single blood sample with no need for prior fasting or glucose ingestion. Currently, an HbA1c of 6.5% (48mmol/mol) is the accepted threshold for the diagnosis of T2DM. Many factors interfere

with HbA1c measurement and interpretation and it is recommended to repeat an HbA1c to exclude laboratory errors in asymptomatic individuals suspected to have T2DM [29]. It is also important to consider factors that affect glycation, such as iron deficiency anemia and to a lesser extent, iron deficiency as well as ethnic differences, when utilizing HbA1c for diagnosis [8, 30].

Population differences in optimal HbA1c thresholds to detect T2DM have, however, been noted and the universal applicability of this diagnostic threshold questioned [31–33]. The role of a single HbA1c measurement to accurately distinguish between women with GDM and overt T2DM in the early postpartum period, remains uncertain [34].

Few global studies have compared the diagnostic utility of HbA1c with OGTT in the early postpartum period, with the predictive value varying depending on the prevalence and degree of hyperglycemia [35–41]. The utility of a single puerperal HbA1c measurement as a diagnostic modality following HFDP has not been established in South African women. It is important to investigate simplified diagnostic methods with appropriate thresholds to streamline postpartum testing in LMICs with limited resources. This study examined the diagnostic utility of HbA1c compared with an OGTT, to identify persistent hyperglycemia 4–12 weeks after delivery, in South African women with HFDP.

## Methods

### Study design and setting

The study followed a descriptive, cross-sectional design. The study population included all women with recent hyperglycemia first detected in pregnancy (HFDP) who attended the postpartum clinic 4–12 weeks after delivery from 1 November 2019 to 31 October 2022. Glucose assessment included an OGTT and HbA1c measurement. Hemoglobin was routinely measured postpartum. Women in this study received prenatal care in the multidisciplinary, diabetes clinic at Tygerberg Hospital, Cape Town, South Africa. As recommended by the World Health Organization, all pregnant women in South Africa are provided with iron and folate supplements to prevent anemia [42, 43]. At Tygerberg, this is also standard practice.

The HFDP diagnosis was based on the criteria outlined in the National Institute of Health Care Excellence (NICE) guidance (fasting plasma glucose (FPG) $\geq$ 5.6mmol/l $\pm$ OGTT 2h-hour plasma glucose (2hPG) $\geq$ 7.8mmol/l). Women who had postpartum hemorrhage, ongoing blood loss or who received a blood transfusion were excluded from study entry. An incomplete OGTT glucose set and a hemoglobin (Hb) concentration $\leq$ 10g/dl also served as exclusion criteria. The research complied with the World Medical Association Declaration of Helsinki ethical principles for medical research involving human subjects and was approved by the Stellenbosch University Health Research and Ethics Committee as well as by local hospital authorities (HREC ref: S21/11/255). Based on the fully anonymized data and standard practice assessment, the ethics committee waived the need for informed consent.

### Data collection

Antenatal clinical information and data were retrieved from the Tygerberg Electronic Content Management System (ECM). Structured, standardized data sheets were used to record clinical and anthropometric data at the early postpartum visit. The assessment was done by dedicated nursing staff, senior medical officers, diabetologists, obstetricians and endocrinologists. Biochemical results were obtained via the National Health Laboratory System (NHLS) electronic platform. Strict confidentiality was maintained as the ECM and NHLS databases, as well as personal computers used for all data collection, are password protected. Patient information and data were entered onto a spreadsheet using Excel version 2019 Microsoft Office

Professional Plus (Microsoft Corp, Redmond, WA, USA). A unique number was linked to each patient's data set and the patient de-identified.

## Demographics and clinical characteristics

A questionnaire, administered at the postpartum visit, included demographic, antenatal and early postpartum information e.g. maternal age and gestation at booking, parity, known family history of diabetes, ethnicity, marital and socio-economic status, the self-reported highest level of education, mode of feeding and personal medical history.

## Anthropometry and blood pressure

Basic anthropometric measurements (weight and height) were taken. Weight was measured with minimal clothing using a manual hospital scale positioned on a flat surface, and recorded to the nearest kilogram (kg). Height was measured without shoes or headwear to the nearest centimetre (cm) with a calibrated stadiometer. Body mass index (BMI) was calculated (kg/m$^2$) and classified according to the World Health Organisation (WHO) global recommendations [44]. Brachial blood pressure was measured with an appropriate cuff size on the left arm while seated, using a calibrated automated sphygmomanometer (Dinamap Carescape V100).

## Biochemistry

The HbA1c test, in addition to a standard 75-gram OGTT (FPG and 2hPG), was performed after an overnight fast. Laboratory analysis was done at the NHLS, a South African National Accreditation System (SANAS) accredited laboratory based at Tygerberg Hospital. NHLS laboratories are accredited by SANAS for compliance with international standards (ISO 15189:2007) [45].

Blood samples were collected in sodium fluoride tubes (Becton Dickenson) for the measurement of plasma glucose using the hexokinase method on the Roche Cobas 6000 (Roche Diagnostics, Mannheim, Germany) platform which has a measuring range from 0.11–41.6 mmol/L with a reported coefficient of variation (CV) of 1.3% at a glucose level of 5.38 mmol/L and 1.1% at a level of 13.4 mmol/L respectively. Internal and external quality controls were performed according to laboratory protocol. The standard turnaround time for glucose samples' (blood sampling to laboratory analysis) was 107 minutes for FPG and 74 minutes for 2hPG. Whole blood collected in potassium-EDTA tubes (Becton Dickinson) was used to measure HbA1c with the turbidimetric inhibition immunoassay on the Tina-quant® HbA1c assay Generation 3 on the Roche Cobas 6000® (Roche Diagnostics, Mannheim, Germany). The measuring range of HbA1c was 4.2%-20.1% (22 to 195mmol/mol), with a CV of 1.3% at HbA1c 5.3% (34mmol/mol) and 1.1% at HbA1c 9.9% (85mmol/mol). The HbA1c determination was done by calculation according to the approved International Federation of Clinical Chemistry(IFCC and Laboratory Medicine method, traceable to the Diabetes Control and Complications Trial. (DCCT) [46]. The HbA1c assay is standardized according to IFCC and transferable to DCCT/NGSP.

## Categorization of study participants

**OGTT (fasting and 2 hour post glucose-load plasma glucose).** Based on the OGTTs, women were categorized according to the American Diabetes Association (ADA) criteria into two main subgroups, namely (i) normoglycemia (FPG <5.6mmol/l and 2hPG <7.8mmol/l _or_ (ii) hyperglycemia (FPG ≥ 5.6mmol/l ± 2hPG ≥ 7.8mmol/l) [29]. Women with hyperglycemia were then further subclassified as (a) prediabetes (impaired fasting glucose (IFG) FPG 5.6–6.9

mmol/l and 2hPG < 7.8mmol/l; impaired glucose tolerance (IGT) FPG < 5.6mmol/l and 2hPG 7.8–11.0mmol/l or both FPG 5.6–6.9mmol/l and 2hPG 7.8–11.0mmol/l) _or_ (b) T2DM (FPG ≥ 7.0 mmol/l ± 2hPG ≥11.0mmol/l). Women with discordant glucose categorization at 0 and 120 mins were allocated according to the highest value.

**HbA1c.** Based on HbA1c, women were also categorized as (i) normoglycemia: HbA1c < 5.7% (< 39 mmol/mol) _or_ (ii) hyperglycemia. Women with hyperglycemia based on HbA1c were then further subclassified as (a) prediabetes (HbA1c ≥ 5.7–6.4% (≥39-47mmol/mol) _or_ (b) T2DM (HbA1c ≥ 6.5% (≥ 48mmol/mol) [29].

## Statistical analyses

The sample size calculation was based on our previous studies with a T2DM prevalence of 25% postpartum and indicated a minimum of 111 subjects. Statistical analysis was performed using GraphPad Prism software (Version 9.5.0). The Shapiro-Wilk and Kolmogorov-Smirnov normality testing was performed to assess data distribution. Normally distributed data are presented as mean ± standard deviations (SD) and non-parametric data as median and interquartile range (IQR). Descriptive statistics were used for socio-demographic parameters. The Wilcoxon matched pairs signed rank test was used to assess differences in parameters between time points (booking vs postpartum). A receiver operator characteristic (ROC) curve was plotted to evaluate the diagnostic performance of HbA1c against the mean glucose derived from the OGTT. Threshold level for significance was accepted at $p < 0.05$. Sensitivity, specificity, negative predictor value (NPV) and positive predictor values (PPV) were calculated as follow: sensitivity = TP/(TP+FN); specificity = TN/(TN+FP); NPV = TN/ (FN+TN); PPV = TP/(TP+FP). TP = true positive; TN = true negative; FP = false positive; FN = false negative.

## Results

In total, 181 women with HFDP attended the early postnatal visit, but fourteen were excluded because glucose datasets were missing in ten cases, Hb- values were below 10g/dL in three cases, and the Hb result was unavailable in one case (see flow sheet; Fig 1). We thus report on the 167 study participants with HFDP who had an OGTT glucose dataset with HbA1c 4–12 weeks postpartum. Participant characteristics are stratified based on postpartum glucose categories in Table 1.

### Antenatal characteristics

Participant characteristics are noted in Table 1. The median age of the total cohort was 34 years (IQR 29–37), and near half of women (73/167; 44%) were of advanced maternal age (≥ 35 years). Most women were multigravidas (89%), either black African (57%) or of mixed ancestry (31%) and educated up to secondary level (78%). Sixty-four of the multigravida women (64/148; 43%) reported at least one previous fetal loss, of which five (5/148; 3%) were in the third trimester.

The median BMI at booking was 38 kg/m$^2$ (IQR 33–45). Individual BMI's, based on WHO categories (overweight BMI 25.0–29.9 kg/m$^2$; obese BMI ≥ 30kg/m$^{2)}$ were overweight or obese in all but five study participants [overweight in 20 (12%), obese in 142 (86%)].

Study participants booked early in the index pregnancy at a median gestation of 12 weeks (IQR 8–17). Only eight women (8/167;5%) had their initial evaluation after 24 weeks. HFDP was diagnosed at a median gestation of 26 weeks (IQR 24–31). At diagnosis, fasting glucose was available in 166/167, 2-hour glucose in 156/167 and HbA1c in 160/167 women, respectively. Forty-six women (46/167; 28%) met the diagnostic criteria for IADPSG overt diabetes. In 41 of these women, a diagnosis of overt diabetes was based on the presence of elevated

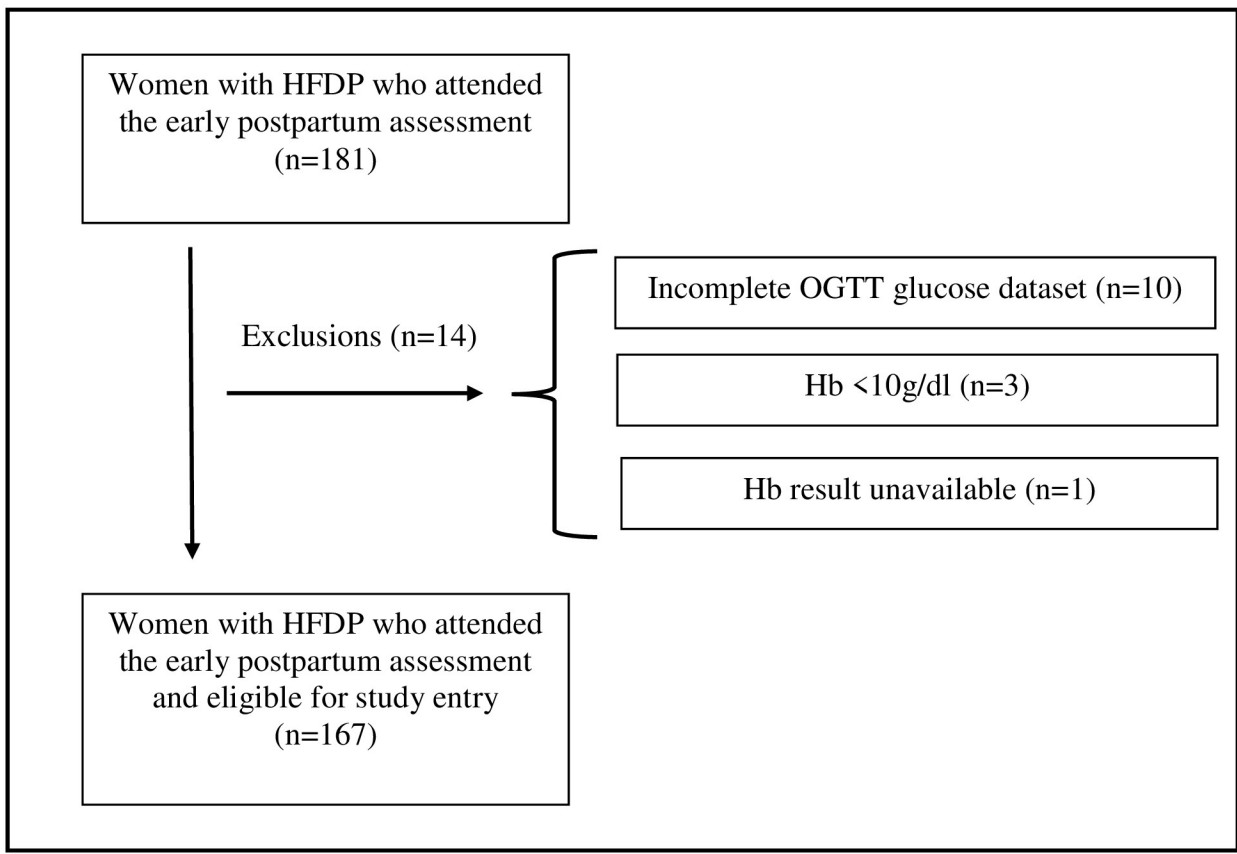

**Fig 1. Flow chart of the study population.** Hb = Haemoglobin, HFDP = hyperglycemia first detected in pregnancy, OGTT = oral glucose tolerance test.

plasma glucose at least one time point during the OGTT. In 22 women both an elevated fasting ($\geq$ 7mmol/l) and an elevated 2-hour value of $\geq$11mmol/l were present, 11 had high fasting glucose only, and eight only an elevated 2-hour value. At diagnosis a HbA1c $\geq$ 6.5% was documented in 34 study participants. Twenty nine of these women had a concomitant diagnostic OGTT criterium for overt diabetes, whereas in five patients the diagnosis was based on a HbA1c only.

Hyperglycemia was managed and glycemic targets were achieved with dietary modifications in 41 (25%); 115 women required additional metformin (69%), and in only a minority of the cohort, both metformin and insulin were needed (11/167; 7%). Third trimester HbA1c was available in more than half (95/167; 56%) of the cohort with a median value of 6% (IQR 5.6–6.5). By using paired analysis to compare the median 3rd trimester HbA1c to the median postpartum HbA1c [5.6 (IQR 5.4–6.2)], the 3rd trimester values were higher by 0.4%.

Delivery information was available for the 164 live births. The median gestation at delivery was 38 weeks (IQR 37–38), while preterm birth (< 37 weeks) occurred in 29 (29/164, 18%). Labour was induced in more than half of the cohort (95/164;58%). This largely reflected the local policy of delivery at 38 weeks' gestation for patients with HFDP on glucose-lowering pharmacotherapy. Normal vaginal delivery (NVD) predominated (100/164; 61%), preceded by induction of labour in most (66/100; 66%). Of the 64 caesarean sections performed, 46 (72%) were emergency procedures for fetal indications Fifteen neonates (15/164; 9%) had macrosomia (>4000g) at birth. Three intra-uterine deaths (IUDs) occurred at 28, 30, and 31 weeks.

**Table 1. Participant characteristics in the total cohort and stratified by postpartum glucose categories.**

|  | Total cohort (n = 167) | Normoglycemic (n = 115) | Prediabetes (n = 34) | T2DM (n = 18) | p-value |
|---|---|---|---|---|---|
| **Age (years) [a]** | 34 (29–37) | 33 (28–37) | 34 (31–37) | 34 (29–36) | n.s |
| • >35 years [n(%)] | 73 (44%) | 49 (43%) | 15 (44%) | 9 (50%) | n.s |
| **Gravidity (n)[a]** | 3 (2–4) | 3 (2–4) | 3 (2–4) | 3 (2–4) | n.s |
| • primigravidas [n(%)] | 19 (11%) | 15 (13%) | 1 (3%) | 3 (17%) | - |
| **Parity (n)[a]** | 3 (2–3) | 2 (2–3) | 3 (2–4) | 3 (2–4) | n.s |
| **Ethnicity [n(%)]** |  |  |  |  |  |
| • Black African | 96 (57%) | 65 (57%) | 20 (59%) | 11 (61%) | n.s |
| • Mixed Ancestry | 52 (31%) | 34 (30%) | 12 (35%) | 6 (33%) | n.s |
| • White | 0 (0%) | 0 (0%) | 0 (0%) | 0 (0%) | - |
| • Indian-Asian Ancestry | 2 (1%) | 1 (1%) | 1 (3%) | 0 (0%) | - |
| • Other | 10 (6%) | 9 (8%) | 0 (0%) | 1 (6%) | - |
| • Not disclosed | 7 (4%) | 6 (5%) | 1 (3%) | 0 (0%) | - |
| **Education level [n(%)]** |  |  |  |  |  |
| • primary | 5 (3%) | 4 (3%) | 0 (0%) | 1 (6%) | - |
| • secondary | 130 (78%) | 85 (74%) | 29 (85%) | 16 (89%) | n.s |
| • tertiary | 24 (14%) | 18 (16%) | 5 (15%) | 1 (6%) | - |
| • not disclosed | 8 (5%) | 8 (7%) | 0 (0%) | 0 (0%) | - |
| **Family history of diabetes[#] [n(%)]** | 72 (43%) | 54 (47%) | 13 (38%) | 5 (28%) | n.s |
| **HIV positive [n(%)]** | 23 (14%) | 18 (16%) | 5 (15%) | 0 (0%) | - |
| **Weight (kg)** |  |  |  |  |  |
| • booking[a] | 96 (83–113) | 93 (81–111) | 104 (86–120) | 97 (89–109) | n.s |
| • 4–12 weeks postpartum[a*] | 91 (80–108) | 91 (79–108) | 98 (85–120) | 98 (87–109) | n.s |
| • weight change | -2.7±5.5 | -2.9±5.8 | -2.1±4.6 | -1.9±5.1 |  |
| **BMI (kg/m$^2$)[a]** |  |  |  |  |  |
| • booking[a] | 38 (33–45) | 38 (33–44) | 38 (33–48) | 37 (33–44) | n.s |
| • 4–12 weeks postpartum[a*] | 36(32–43) | 36(31–42) | 36(32–46) | 37(33–42) | n.s |
| **Hypertension** |  |  |  |  |  |
| • chronic [n(%)] | 29 (17%) | 19 (17%) | 5 (15%) | 5 (28%) | n.s |
| • new onset gestational [n(%)] | 30 (18%) | 20 (17%) | 4 (12%) | 5 (28%) | - |
| **Gestation at diagnosis (weeks)[a]** | 26 (24–31) | 27 (25–31) | 25 (22–28) | 26 (22–29) | Normal vs pre (p = 0.0211) |

Unless otherwise specified, values are given as number (%) or median and interquartile range[a].

[#]Family history of diabetes refers to a first-degree relative with diabetes.

[*]p-value < 0.001 in comparison to booking measurement.

BMI = Body Mass Index in kg/m$^2$. IADPSG T2DM = International Association for Diabetes in Pregnancy Study Group (IADPSG) criteria for Type 2 Diabetes Mellitus (T2DM) (fasting plasma glucose of ≥7mmol/l and/or 2 hours post 75gram glucose value (2hPG) ≥ 11.1mmol/l) [29].

Two of these occurred during maternal admission with severe diabetic ketoacidosis (DKA) (glucose and pH values of 21.5mmol/l and 7.1; 27mmol/l and 7.11, respectively). Preeclampsia and a urinary tract infection precipitated these DKA's.

## Postpartum assessment

After delivery, the median number of weeks to postpartum OGTT was 10 (IQR 7–12). Pairwise analysis indicated a significant decrease (p<0.001) in both weight and BMI (kg/m$^2$) at the time of the postpartum assessment compared to booking, with the largest weight change observed in the postpartum normoglycemic group (Table 1). Two-thirds of women (112/167; 67%) reported exclusive breastfeeding.

**Table 2. Antenatal and postpartum parameters of glucose homeostasis within OGTT postpartum glucose categories (n = 167).**

| Postpartum OGTT category | Antenatal glycemic parameters | Postpartum glycemic parameters | P value |
|---|---|---|---|
| **Normoglycemia n = 115 (69%)** | | | |
| • FPG mmol/l[a] | 5.5 (4.7–6.1) | 4.7 (4.4–5) | p<0.001 |
| • 2hPG mmol/l[a] | 8.7 (8.1–9.5) | 6.1 (4.9–6.7) | p<0.001 |
| • HbA1c %[a] | 5.7 (5.3–6.0) | 5.5 (5.3–5.8) | p = 0.003 |
| • *Concordant HbA1c classification* | | *77/115 (67%)* | |
| **Hyperglycemia n = 52 (31%)** | | | |
| • *Concordant HbA1c classification* | | *44/52 (85%)* | |
| **Prediabetes n = 34 (20%)** | | | |
| • FPG mmol/l[a] | 6.2 (5.2–8.4) | 6.0 (5.3–6.4) | p = 0.0222 |
| • 2hPG mmol/l[a] | 9.6 (8.7–10.9) | 8.5 (6.7–9.3) | p<0.001 |
| • HbA1c %[a] | 6.1 (5.9–7.1) | 6.3 (5.8–6.8) | n.s |
| • *Concordant HbA1c classification* | | *15/34 (44%)* | |
| • *HbA1c ≥6.5%* | | *12/34 (35%)* | |
| **T2DM n = 18 (11%)** | | | |
| • FPG mmol/l[a] | 6.4 (5.7–8.2) | 7.3 (6.9–7.9) | n.s |
| • 2hPG mmol/l[a] | 9.2 (7.0–11.7) | 12 (11.5–14.5) | n.s |
| • HbA1c %[a] | 6.6 (5.9–7.3) | 6.8 (6.5–7.2) | n.s |
| • *Concordant HbA1c classification* | | *15/18 (83%)* | |

Values are given in median IQR[a] or number (%) FPG = Fasting plasma glucose. 2hPG = Plasma glucose 2hrs after 75-gram OGTT. HbA1c = glycosylated haemoglobin. HbA1c classification: normoglycemia: HbA1c < 5.7%(39mmol/mol); hyperglycemia: HbA1c ≥5.7; prediabetes: HbA1c 5.7–6.4%(≥39-47mmol/mol); T2DM: HbA1c ≥6.5% (≥48mmol/mol).

Table 2 provides information on the antenatal and postpartum biochemical parameters of glucose homeostasis stratified according to postpartum OGTT defined glucose subcategories. Median (IQR) fasting plasma glucose (FPG), 2-hour post-OGTT plasma glucose (2hPG) and plasma HbA1c are tabulated. Concordance with HbA1c categorization is also demonstrated. The postpartum OGTT indicated hyperglycemia in 52/167 (31%) participants. Thirty-four women (20%) had prediabetes and 18 (11%) had T2DM. In the prediabetes category, both FPG *and* 2hPG values were diagnostic in 12 patients, whereas a third of women (11/34; 32%) presented with an isolated abnormal FPG or an abnormal 2hPG respectively. Of the 18 women with T2DM, both FPG *and* 2hPG values were diagnostic in six patients, the diagnosis was based on FPG only in eight patients, and in four patients only an abnormal 2hPG was present. Based on fasting glucose only, 15 women with persistent hyperglycemia (11 with prediabetes and four with T2DM) in our cohort would not have been identified (15/52; 29%). An HbA1c correctly classified 85% of the 52 participants with gold standard OGTT defined hyperglycemia and 15 of 18 women with postpartum persistent T2DM.

The sensitivity, specificity, positive and negative predictive values to correctly identify normoglycemia and specific subcategories of hyperglycemia based on OGTT criteria, are detailed in S1 Fig. At the early postpartum visit, our OGTT classified 115 of our total cohort as normoglycemic. In the OGTT normoglycemic category, HbA1c indicated hyperglycemia in 38/115 (33%), 35 with prediabetes and three with diabetes. The sensitivity and specificity of a single HbA1c postpartum to correctly identify the degree of hyperglycemia compared to the OGTT were best for T2DM (sensitivity 83%; specificity 97%).

The receiver operating characteristic (ROC) curves for the diagnostic performance of HbA1c and FPG against OGTT are presented in Fig 2. HbA1c had a higher sensitivity to detect

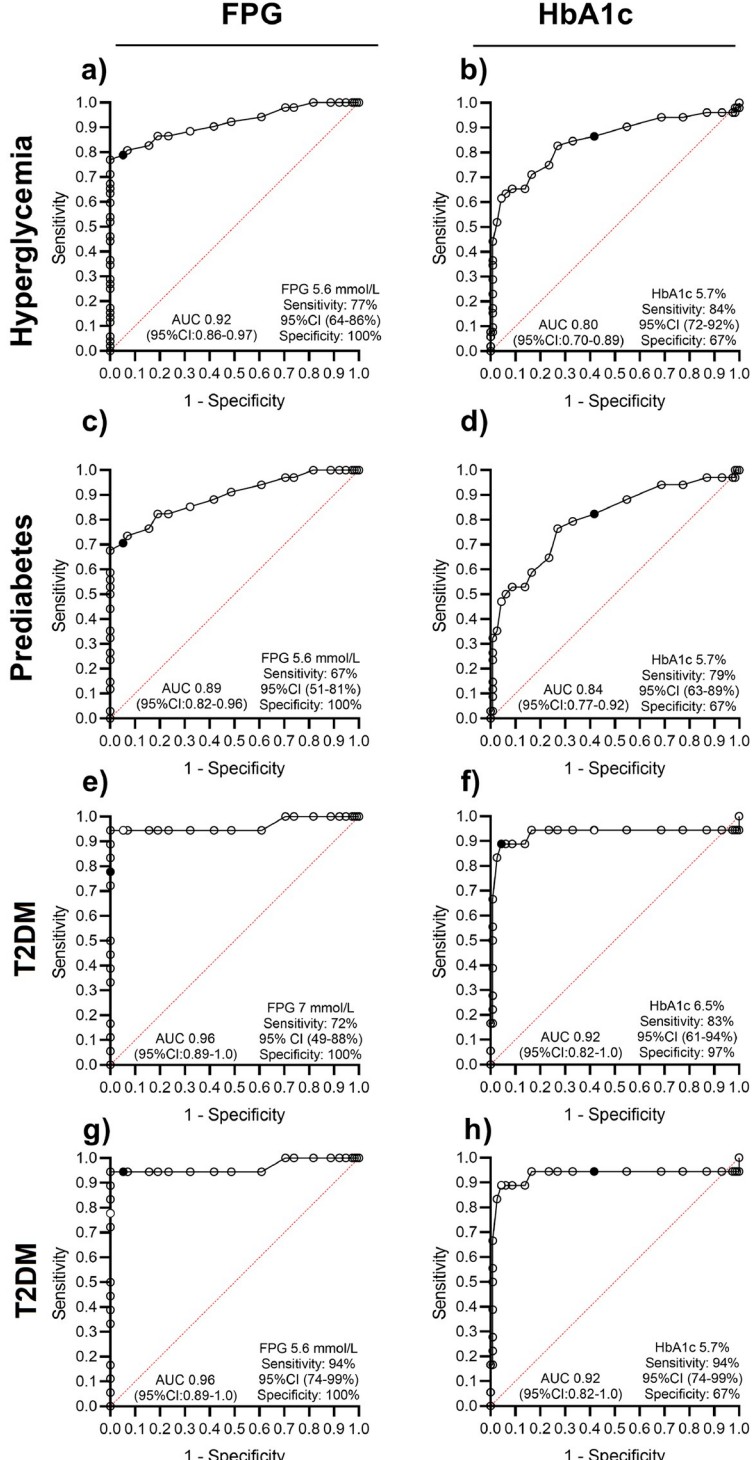

**Fig 2. ROC curves comparing HbA1c to OGTT within diagnostic glucose categories.**

hyperglycemia than FPG at conventional diagnostic thresholds in all three glucose categories (Fig 2a–2f). By lowering the HbA1c threshold to ≥5.7% (39mmol/mol) for T2DM, the sensitivity improved to 94% [AUC 0.92 (95% CI 0.82–1.0)], but significantly worsened specificity to 67% (Fig 2h).

## Discussion

Although not ideal, a single postpartum HbA1c at conventional diagnostic thresholds is clinically useful in detecting hyperglycemia in women with prior HFDP. The ability of a single HbA1c to differentiate between the two hyperglycemia classes, i.e., prediabetes and T2DM, was limited, with the test more accurate in predicting T2DM. It is noteworthy that 33% of patients with OGTT normoglycemia had an HbA1c measurement of ≥ 5.7% (39mmol/mol) and were therefore over diagnosed. ROC curves based on the gold standard postpartum OGTT indicated that HbA1c was comparable as a diagnostic test but did not outperform the ability of fasting blood glucose to detect hyperglycemia. The sensitivity of HbA1c improved to an ideal 94% when the conventional threshold for T2DM was adjusted to the prediabetes cut-off [HbA1c ≥ 5.7% (39mmol/mol)]. This, however, came at the expense of a significant decline in specificity from 97% (conventional threshold) to 63% (lowered threshold).

In pregnancies complicated by HFDP, pragmatic solutions to optimally categorize and diagnose glucose homeostasis early postpartum are especially valuable in resource-limited environments. Identifying persistent hyperglycemia postpartum is important, as timeous pharmacological intervention in those with T2DM has significant long-term health benefits. HbA1c represents an attractive diagnostic alternative to a postpartum OGTT. The test can be performed without any patient-related restrictions, in contrast to obligatory fasting with an FPG or when performing an OGTT and is thus ideal in the out-patient setting.

Non-communicable diseases associated with obesity are among the top causes of death in South Africa [47]. Globally, the rising prevalence of T2DM parallels that of obesity and South African women of childbearing age have a disproportionately high burden [6]. Recent estimations noted an escalation in the prevalence of overweight and obesity of 60% and 35.2%, respectively, from 1998 and 2017 [6]. A concerning 97% of women in our study cohort were overweight or obese. T2DM, prediabetes and obesity lead to several maternal and child health problems extending far beyond pregnancy. The risk of developing T2DM in patients with prediabetes varies and is influenced by the degree of hyperglycemia, as well as the individual background genetics and pancreatic β-cell reserve. Individuals with both IFG and IGT, develop T2DM sooner than those with isolated defects [48]. In our study, postpartum HbA1c overestimated the number of women with T2DM, incorrectly classifying 12 women with OGTT prediabetes and three with OGTT normoglycemia as having T2DM, based on OGTT criteria. Most (7/12) of the women diagnosed with OGTT prediabetes had both IFG and IGT on OGTT. It can thus be argued that HbA1c tends to overestimate the presence of T2DM, mostly in a subset of women with prediabetes at highest risk for future T2DM.

The landmark randomised controlled trial "The Diabetes Prevention Program Outcomes Study" indicated that intensive lifestyle modification and metformin are highly effective in delaying the progression to T2DM in women with a history of GDM [24, 25]. Women with OGTT prediabetes and a HbA1c ≥ 6.5 (48mmol/mol) represent a subset at higher risk within the prediabetes range and may benefit from earlier pharmacological intervention with metformin.

Fasting plasma glucose, an OGTT and HbA1c are well-established screening tools for T2DM. FPG and HbA1c are convenient, but the diagnostic accuracy varies according to cut-

offs, the background diabetes prevalence in the population and ethnicity. In this study, an early postpartum HbA1c ≥ 6.5% (48mmol/mol) had a sensitivity of 83% to detect T2DM based on OGTT glucose criteria, identifying 15/18 women. In comparison a FPG of ≥ 7mmol/L alone had a sensitivity of 72% to detect T2DM and missed four of eighteen women with T2DM. Our small study thus indicated that the sensitivity of a HbA1c was ±20% better compared to a diagnostic FPG in T2DM. Chivese and colleagues assessed the diagnostic accuracy of HbA1c vs FPG and OGTT in non-pregnant individuals from seven African countries [49]. The ethnic backgrounds resembled the ethnic representation in our cohort and included predominantly Black African women and women of Mixed Ancestry. In their study, the sensitivity of a HbA1c ≥ 6.5% (48mmol/mol) to diagnose T2DM was 58% vs OGTT, a value significantly lower than the 83% documented in our cohort. As expected, sensitivity improved by lowering the cut off [HbA1c ≥ 6% (42mmol/mol)] in their study, but this came at the expense of decreased specificity. Another recent study suggested that an HbA1c cut-off of 6.5% (48mmol/mol) may miss up to 50% of non-pregnant persons with T2DM, perhaps attributable to ethnic and population differences [50]. This study suggests an HbA1c ≥6.1% may be more appropriate in the mixed ancestry population.

The conventional HbA1c threshold may thus not be globally applicable and may be too high in certain populations. This might also apply to postpartum assessment following pregnancies complicated by HFDP. The reported diagnostic accuracy of HbA1c for T2DM postpartum is impacted by the antenatal diagnostic criteria used, the timing of the early postpartum OGTT and the background T2DM prevalence. Katreddy evaluated 203 predominantly Caucasian women ± six weeks after delivery and reported a sensitivity of 71% and 99% specificity for T2DM with the conventional HbA1c threshold of 6.5% (48mmol/mol). By using an HbA1c of 6.0% (44mmol/mol), sensitivity improved to 100% [area under the curve: 0.98 (95%CI: 0.96–1.00)] [39]. Kim et al. evaluated 699 Korean women with GDM, 6–12 weeks postpartum. The prevalence of T2DM and prediabetes was 5.2% and 49.1% respectively [40]. They found that a HbA1c was not sensitive enough for early postpartum testing with poor sensitivity to detect both T2DM and prediabetes in their cohort of women (19.4% for T2DM and 41.2% for prediabetes). The authors speculated that the usefulness and diagnostic sensitivity of HbA1c might improve with the addition of a FPG [41]. In a study by Picón et al. that included 231 Spanish women with prior GDM, the ability of a HbA1c or a FPG alone or a HbA1c combined with a FPG to identify women with persistent hyperglycemia based on OGTT criteria was evaluated. The OGTT identified 39% women with prediabetes and 6% with diabetes. A HbA1c of ≥ 6.5% (48mmol/mol) identified 19% with prediabetes and only one woman with T2DM. The combination of HbA1c and FPG did not improve the sensitivity to detect hyperglycemia compared to a FPG only. Claesson evaluated 140 women 1–5 years after delivery and combined FPG with HbA1c, but also found that the combination did not improve the performance of FPG per se [35].

In the study cohort, the yield to identify T2DM increased significantly (sensitivity 94%) when the HbA1c diagnostic threshold was adjusted to ≥ 5.7% (39mmol/mol). The sensitivity did not improve with the addition of a FPG ≥ 5.6mmol/l, but the specificity of the FPG per se was better (100% vs 67%). In the prediabetes category, a HbA1c ≥ 5.7% (39mmol/mol) outperformed FPG ≥ 5.6mmol/l as a stand-alone test, with a sensitivity of 79% for HbA1c compared to 67% for FPG.

There are several plausible reasons, for the variation in the optimal diagnostic set points of HbA1c and glucose. The concentration of plasma glucose ex-vivo decreases due to glycolysis [51]. To prevent glycolysis, fluoride tubes are used in clinical practice and were utilized in this study. Complete inhibition of glycolysis can take as long as four hours, with a resultant potential decline in measured glucose up to 0.6mmol/mol. Alternatively, glycolysis can be

inhibited by separating plasma from cells within 30 minutes after venesection and transporting samples in ice slurries [51]. This is mainly used in research settings, as it is impractical in clinical practice. In our study, the reported turnaround times (74–107 minutes) from sampling to analysis were relatively short, and it is thus unlikely that glycolysis significantly impacted glucose measurements. Breastfeeding has short and long-term benefits in women with GDM, including delaying the onset of T2DM by impacting insulin sensitivity. Breastfeeding should thus be encouraged [52–54]. Most (two-thirds) women in the study were breastfeeding at the time of the assessment, raising the question of whether this may have influenced or modified glucose responses to an OGTT. A randomized crossover trial investigated the impact of breastfeeding on OGTT in women with recent GDM (±12 weeks postpartum) and concluded that breastfeeding did not impact postpartum glucose categorization [55]. HbA1c as a monitoring tool reflects the mean plasma glucose over ±120 days (the lifespan of the red blood cells) in non-pregnancy. This relationship may be affected by conditions that alter red blood cell turnover, including iron-deficient states and pregnancy per se [30, 40, 51, 56]. In iron-deficient states, HbA1c is falsely elevated, but declines by 0.4% after one month's supplementation [30]. Consequently, one must consider that HbA1c levels in the study could have been altered, the direction dependent on the timing of iron supplementation. Iron supplementation is initiated routinely at the first antenatal visit in our setting. Most women booked early and before 24 weeks gestation, making a variation in the measured HbA1c postpartum due to iron supplementation unlikely. HbA1c, in contrast to the impact of iron deficiency, may be lowered due to physiological changes of pregnancy per se. The increased red cell turnover in pregnancy reduces glycation of hemoglobin and can lower HbA1c by as much as 0.5% in the third trimester [57]. Moreover, evidence indicates a delay between the development of hyperglycemia and increasing HbA1c [58]. As a result, postpartum HbA1c may still reflect a proportion of third-trimester glycemia. In our study, the postpartum assessment occurred ten weeks (IQR 7–12) after delivery. The 3rd trimester HbA1c was higher than postpartum and antenatal representation are thus excluded. At present, HbA1c results are interpreted based on the assumption that erythrocyte life span is constant at 120 days when, in fact, enormous heterogeneity occurs due to biological variation [59]. Possibly this was also a contributing factor in our study, explaining the lack of correlation between 3rd trimester and postpartum HbA1c. In view of the fact that our analysis did not include individual indices of red blood cell survival, we did not take into account such heterogeneity. It may be beneficial for future research to assess red cell survival, particularly in early postpartum assessments.

This is the first study in South-African women with recent HFDP where the diagnostic accuracy of a single HbA1c is compared with the gold standard OGTT early postpartum. The study had limitations. It is not a large study, antenatal data were gathered retrospectively and there was no control group. Population-specific thresholds associated with adverse outcomes in T2DM and prediabetes and the ability of postpartum HbA1c to predict future T2DM warrants further investigation as it may inform required long-term follow-up intervals. The various pathophysiological contributors to abnormal glucose homeostasis and their relation to postpartum HbA1c are also worthy of further interrogation as are tests evaluating glycemia not influenced by red blood cell kinetics such as glycated albumin.

In summary, good sensitivity of an early postpartum HbA1c to detect women with established T2DM and those at the highest risk for future T2DM (prediabetes with combined FPG and IGT) is demonstrated in this study. These women represent at risk individuals who will benefit from early intervention. In clinical practice, HbA1c is valuable when obligatory fasting and other pre-analytical OGTT testing requirements are not feasible but cannot unequivocally replace OGTT based on this study. Further evaluation in a larger cohort is required.

## Supporting information

**S1 Fig. The performance discordance of Hba1c categorisation compared to oral glucose tolerance testing.**
(TIF)

## Author Contributions

**Conceptualization:** Ankia Coetzee, Magda Conradie.

**Data curation:** Ankia Coetzee, Magda Conradie.

**Formal analysis:** Ankia Coetzee, Mari van de Vyver, Magda Conradie.

**Investigation:** Ankia Coetzee.

**Methodology:** Ankia Coetzee, David R. Hall, Mari van de Vyver.

**Project administration:** Ankia Coetzee.

**Software:** Mari van de Vyver.

**Supervision:** David R. Hall, Magda Conradie.

**Validation:** Ankia Coetzee, Mari van de Vyver.

**Visualization:** Ankia Coetzee.

**Writing – original draft:** Ankia Coetzee, David R. Hall, Mari van de Vyver, Magda Conradie.

**Writing – review & editing:** Ankia Coetzee, David R. Hall, Mari van de Vyver, Magda Conradie.

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
