## [Decision Letter · Decision Letter 0]

28 Mar 2023

PONE-D-23-03908Early postpartum HbA1c after hyperglycemia first detected in pregnancy - imperfect but not without valuePLOS ONE

Dear Dr. Coetzee,

Thank you for submitting your manuscript to PLOS ONE. After careful consideration, we feel that it has merit but does not fully meet PLOS ONE’s publication criteria as it currently stands. Therefore, we invite you to submit a revised version of the manuscript that addresses the points raised during the review process.

We look forward to receiving your revised manuscript.

Kind regards,

Jaya Anna George, BMBCH

Academic Editor

PLOS ONE

Journal Requirements:

Additional Editor Comments:

Dear Dr Coetzee,

Thank you for this interesting and important study. The reviews are attached. Please address these and prepare a point by point rebuttal and resubmit.

Kind Regards,

Jaya George

Reviewers' comments:

Reviewer's Responses to Questions

**Comments to the Author**

1. Is the manuscript technically sound, and do the data support the conclusions?

Reviewer #1: Yes

Reviewer #2: Yes

2. Has the statistical analysis been performed appropriately and rigorously? 

Reviewer #1: Yes

Reviewer #2: Yes

3. Have the authors made all data underlying the findings in their manuscript fully available?

Reviewer #1: Yes

Reviewer #2: Yes

4. Is the manuscript presented in an intelligible fashion and written in standard English?

Reviewer #1: Yes

Reviewer #2: Yes

5. Review Comments to the Author

Reviewer #1: Thank you for the opportunity to review the manuscript “Early postpartum HbA1c after hyperglycemia first detected in pregnancy – imperfect but not without value”

This is an interesting novel study and should be published. However, some revisions are needed before this manuscript can be considered.

Keywords:

These are too long- consider following MeSH terms: hyperglycemia, pregnancy, gestational diabetes, HbA1c, postpartum, oral glucose tolerance test

Abstract:

- Results confusing – consider rewriting clarifying that hyperglycemia consists of prediabetes and diabetes. Maybe instead of using term “hyperglycemia”, divide into normoglycemia, prediabetes and diabetes?

Introduction:

- Generally well written.

- However, there is one MAJOR error that authors need to correct before this can be published: in the introduction, authors refer to “glycosylated” hemoglobin and in the discussion they mention “glycosylation”. GLYCOSYLATION is an enzymatic process, catalysed by glycosyltransferase, where a carbohydrate group is added to a protein. This is important for protein folding and activity, cell-to-cell adhesion, diversity of the proteome, etc and is a non-pathological process. However, GLYCATION refers to the non-enzymatic addition of a carbohydrate to a protein, is a pathological process and associated with glycaemia and aging. Many old papers refer to HbA1c as “Glycosylated haemoglobin” which is incorrect. Please correct introduction to “glycated haemoglobin” and discussion to “glycation”!

- HbA1c > 6.5% should be repeated for diagnosis of DM

- Non-glycemic factors affecting HbA1c should be mentioned e.g. iron status, illness, genetic and haematological factors and ethnicity

Methodology:

- Why was iron status / iron treatment not considered? Both common in pregnancy affect HbA1c.

- Four weeks postpartum may be too soon as may still reflect pregnancy HbA1c – should have been determined at 12 weeks minimum (red blood cell life span of 120 days)

- Authors should mention that lab accredited to ISO 15189 (2012) standards

- What instrument is glucose determined on?

- Should mention that internal and external quality control performed.

- Correct HbA1c method: Tina-quant ® HbA1c assay Generation 3 on the Roche Cobas 6000 ® (Roche Diagnostics, Mannheim, Germany). This assay is standardized according to IFCC and transferable to DCCT/NGSP.

Results

- Well-written but quite long

- P-values should be given for comparisons

Discussion:

- Generally well written

- Start discussion by listing key findings of the study without actually reseating the results

- Some areas need to be clarified e.g. page 19 mentions that “women with prediabetes and HbA1c >- 6.5% at higher risk for T2DM. Isn’t it DM with that HbA1c?

Tables:

- P-values should be given for comparisons

Minor:

- Check consistency – authors using American English, yet haemoglobin spelt “haemoglobin” in introduction.

- NHLS does not need to be written out in full again under Biochemistry

- ADA in methodology needs to be written out

- HbA1c subheading in methodology – small c

- Don’t write out T2DM in full again in discussion

Reviewer #2: A study from Tygerberg has shown that an HbA1c of >=6.1 may be more appropriate in the mixed ancestry population.

Did you look at this. HbA1c has several limitations in our populations such as the fact that it is affected by HIV , iron deficiency.

This should be mentioned

6. PLOS authors have the option to publish the peer review history of their article (what does this mean?). If published, this will include your full peer review and any attached files.

Reviewer #1: No

Reviewer #2: No

---

## [Author Response · Author response to Decision Letter 0]

10 May 2023

Dear Sir/Madam

Please accept our sincere thanks for providing us with the opportunity to revise our paper “Early postpartum HbA1c after hyperglycemia first detected in pregnancy - imperfect but not without value.”

It was appreciated that excellent recommendations and comments were provided, which, once addressed, improved the rigor of the paper. We would like to express our sincere appreciation to Reviewer 1 for their thoroughness in conducting the review. 

Listed below are the revisions (italics) made to the manuscript per reviewer. To draw the attention of Reviewer 2, yellow highlights have been added to the existing sections. Thus, we were able to avoid duplication.

Yours sincerely

Dr Ankia Coetzee 

Reviewer 1

This is an interesting novel study and should be published. However, some revisions are needed before

this manuscript can be considered.

1. Keywords:

These are too long- consider following MeSH terms: hyperglycemia, pregnancy, gestational diabetes,

HbA1c, postpartum, oral glucose tolerance test

Replaced with:

Keywords: Hyperglycemia, pregnancy, gestational diabetes mellitus, oral glucose tolerance test, HbA1c, postpartum assessment.

2. Abstract:

- Results confusing – consider rewriting clarifying that hyperglycemia consists of prediabetes and

diabetes. Maybe instead of using term “hyperglycemia”, divide into normoglycemia, prediabetes

and diabetes?

Thanks, replaced with:

Results: Glucose homeostasis was assessed at 10 weeks (IQR 7-12) after delivery. Of the 167 participants, 52 (31%) had hyperglycemia, which was comprised of 34 (20%) prediabetes and 18 (11%) T2DM. Twelve women in the prediabetes subgroup had diagnostic fasting plasma glucose (FPG) and 2-hour plasma glucose (2hPG), but in two-thirds of the patients (22/34) only one time point proved diagnostic. The FPGs and the 2hPGs of six women with HbA1c-based T2DM were both within the prediabetes diagnostic range. According to the HbA1c measurements, 85% of 52 participants with gold standard OGTT defined hyperglycemia (prediabetes and T2DM) as well as 15 of 18 women with postpartum persistent T2DM were correctly classified. According to FPG, 15 women with persistent hyperglycemia would have been missed (11 with prediabetes and four with T2DM; 29%).When compared to an OGTT, a single HbA1c of 6.5% (48mmol/mol) postpartum demonstrated a sensitivity of 83% and specificity of 97% for the identification of T2DM. 

3.Introduction:

- Generally well written.

a)GLYCOSYLATION

However, there is one MAJOR error that authors need to correct before this can be published: in the introduction, authors refer to “glycosylated” hemoglobin and in the discussion they mention “glycosylation”. GLYCOSYLATION is an enzymatic process, catalysed by glycosyltransferase, where a carbohydrate group is added to a protein. This is important for protein folding and activity, cell-to-cell adhesion, diversity of the proteome, etc and is a non-pathological process. However, GLYCATION refers to the non-enzymatic addition of a carbohydrate to a protein, is a pathological process and associated with glycaemia and aging. Many old papers refer to HbA1c as “Glycosylated haemoglobin” which is incorrect. Please correct introduction to “glycated haemoglobin” and discussion to “glycation”!

I have addressed the matter, thank you very much.

b) HbA1c > 6.5% should be repeated for diagnosis of DM

A modification was made to the sentence to reflect the suggestion of the reviewers

Currently, an HbA1c of 6.5% (48mmol/mol) is the accepted threshold for the diagnosis of T2DM, however it is recommended to repeat an HbA1c to exclude laboratory errors in asymptomatic individuals.

c) Non-glycemic factors affecting HbA1c should be mentioned e.g. iron status, illness, genetic and

haematological factors and ethnicity

It has been added that:

It is also important to consider factors that affect glycation, such as iron deficiency anemia and to a lesser extent, iron deficiency as well as ethnic differences, when utilizing HbA1c for diagnosis.

4.Methodology:

a) Why was iron status / iron treatment not considered? Both common in pregnancy affect HbA1c.

• We note and agree that iron deficiency and iron deficiency anemia may affect HbA1c levels. This study excluded women with anemia, so the following additional information on iron deficiency has been added to methods:

As recommended by the World Health Organization, all pregnant women in South Africa are provided with iron and folate supplements to prevent anemia.(42,43) At Tygerberg, this is also standard practice at the first antenatal visit.

• Based on our data, our cohort's first prenatal visit occurred at a median gestation of 10 weeks, and delivery occurred at a median gestation of 38 weeks. Based on standard practice, these women would have been supplemented from 10 weeks until 38 weeks of pregnancy. Considering the interval between replacement and assessment in this study, iron deficiency probably would not have an impact on HbA1c levels. In the manuscript's discussion section, this topic is also addressed. The extract is below.

“In iron-deficient states, HbA1c is falsely elevated, but declines by 0.4% after one month's supplementation.(30) Consequently, one must consider that HbA1c levels in the study could have been altered, the direction dependent on the timing of iron supplementation. Iron supplementation is initiated routinely at the first antenatal visit in our setting. Most women booked early and before 24 weeks gestation, making a variation in the measured HbA1c postpartum due to iron supplementation unlikely.” 

b) Four weeks postpartum may be too soon as may still reflect pregnancy HbA1c – should have been determined at 12 weeks minimum (red blood cell life span of 120 days).

Sure, that is an excellent point. The timing of HbA1c testing is also relevant from a therapeutic standpoint, as it is determined in accordance with the historical lifespan of red blood cells (~90 to 120 days). In the following paragraphs, we provide an explanation from a clinical and biochemical perspective. 

Clinical perspective

The postpartum OGTT is widely regarded as a valuable addition to women's preventative health care. However, postpartum attendance rates are low, and dropout rates increase over time. 

For the first postpartum OGTT in women with GDM, the American Diabetes Association and the American College of Obstetricians and Gynecologists recommend a period 4 to 12 weeks. Interestingly, the timing of the first scheduled vaccination of newborns dictated the selection of this window. Generally, clinical settings adhere to this practice for convenience and in order to retain women who are at greater risk. This is also standard practice at our institution. 

Thus, we continue to use this model to provide clinically and contextually relevant information as well as to enable a broader application of postpartum testing in the future.

Biochemical perspective

Discrepancies between HbA1c and newer glucose measurements are increasingly noted in light of technological advancements and increased availability of, for example, continuous glucose monitoring (CGM). HbA1c results are currently interpreted based on the assumption that erythrocyte lifespan is constant, which is 120 days, when in fact, it has a wide distribution (106 ± 21 days). The average exposure time of hemoglobin to glucose is, therefore, subject to significant biological variations (CVx 20%) that are greater than the analytical error of modern assays. For example, in Cohen's study, red cell survival ranged from 39 to 56 days among diabetics and 38 to 60 days among non-diabetics. Glycated albumin (GA) and labile HbA1c (lHbA1c) have therefore been the subject of increased research, since they provide more information about recent intervals. It would have been a great addition to our study if CGM, GA and lHbA1c had been available for comparison.

The reviewer and we agree that postpartum HbA1c may still represent a proportion of third trimester (lower HbA1c) glucose levels. It has been argued that maternal glycemia during the third trimester is often underestimated, as HbA1c levels are lower during this period, but our study revealed that third trimester HbA1c was generally higher than postpartum HbA1c albeit with no correlation. We elaborated on this possibility in the discussion section of our manuscript. The extract reads: “In our study, the postpartum assessment occurred ten weeks (IQR 7-12) after delivery. The 3rd trimester HbA1c was higher than postpartum and antenatal representation are thus excluded”. 

Since the postpartum OGTT was conducted closer to 12 weeks (10 weeks (IQR 7-12)), there can be some comfort in the results but our study did not assess red cell survival.

In the discussion section, the following limitation has been added: "At present, HbA1c results are interpreted based on the assumption that erythrocyte life span is constant at 120 days when, in fact, enormous heterogeneity is a result of biological variation. Possibly this was also a contributing factor in our study, explaining the lack of correlation between 3rd trimester and postpartum HbA1c. In view of the fact that our analysis did not include individual indices of red blood cell survival, we did not take into account such heterogeneity. It may be beneficial for future research to assess red cell survival, and this will be particularly important in early postpartum assessments.”

References:

1) Beck RW, Connor CG, Mullen DM, Wesley DM, Bergenstal RM. The fallacy of average: how using HbA1c alone to assess glycemic control can be misleading. Diabetes care. 2017 Aug 1;40(8):994-9.

2) Malka R, Nathan DM, Higgins JM. Mechanistic modeling of hemoglobin glycation and red blood cell kinetics enables personalized diabetes monitoring. Science translational medicine. 2016 Oct 5;8(359):359ra130

3) Kaiafa G, Veneti S, Polychronopoulos G, Pilalas D, Daios S, Kanellos I, Didangelos T, Pagoni S, Savopoulos C. Is HbA1c an ideal biomarker of well-controlled diabetes?. Postgraduate Medical Journal. 2021 Jun;97(1148):380-3.

4) Cohen RM, Franco RS, Khera PK, Smith EP, Lindsell CJ, Ciraolo PJ, Palascak MB, Joiner CH. Red cell life span heterogeneity in hematologically normal people is sufficient to alter HbA1c. Blood. 2008 Nov 15;112(10):4284-91. doi: 10.1182/blood-2008-04-154112. Epub 2008 Aug 11. PMID: 18694998; PMCID: PMC2581997.

5) Delanghe JR, Lambrecht S, Fiers T, Speeckaert MM. Labile glycated hemoglobin: an underestimated laboratory marker of short term glycemia. Clinical Chemistry and Laboratory Medicine (CCLM). 2022 Feb 1;60(3):451-5.

c) Authors should mention that lab accredited to ISO 15189 (2012) standards

The section on NHLS and SANAS was expanded by adding: NHLS laboratories are accredited by SANAS for compliance with international standards (ISO 15189:2007).(45)

d) What instrument is glucose determined on? Should mention that internal and external quality control performed.

Glucose testing were on the Roche Cobas 6000 (Roche Diagnostics, Mannheim, Germany) platform. The instrument was added to the methods that now reads as follows:

“Blood samples were collected in sodium fluoride tubes (Becton Dickenson) for the measurement of plasma glucose using the hexokinase method on the Roche Cobas 6000 (Roche Diagnostics, Mannheim, Germany) platform which has a measuring range from 0.11–41.6 mmol/L with a reported coefficient of variation (CV) of 1.3% at a glucose level of 5.38 mmol/L and 1.1% at a level of 13.4 mmol/L respectively Internal and external quality controls were performed according to laboratory protocol.”

e) Correct HbA1c method: Tina-quant ® HbA1c assay Generation 3 on the Roche Cobas 6000 ® (Roche Diagnostics, Mannheim, Germany). This assay is standardized according to IFCC and transferable to DCCT/NGSP.

Changed the text to align it. It now reads: 

“Whole blood collected in potassium-EDTA tubes (Becton Dickinson) was used to measure HbA1c with the turbidimetric inhibition immunoassay on the Tina-quant ® HbA1c assay Generation 3 on the Roche Cobas 6000 ® (Roche Diagnostics, Mannheim, Germany). The measuring range of HbA1c was 4.2%-20.1% (22 to 195mmol/mol), with a CV of 1.3% at HbA1c 5.3% (34mmol/mol) and 1.1% at HbA1c 9.9% (85mmol/mol). The HbA1c determination was done by calculation according to the approved International Federation of Clinical Chemistry (IFCC) and Laboratory Medicine method, traceable to the Diabetes Control and Complications Trial. (DCCT).(46) The HbA1c assay is standardized according to IFCC and transferable to DCCT/NGSP”.

Results

- Well-written but quite long

- P-values should be given for comparisons @Magda – I cannot find any comparisons, please will you double check?

Discussion:

- Generally well written

- Start discussion by listing key findings of the study without actually reseating the results

The current start of the discussion” Our study data documented that a single postpartum HbA1c, although not ideal, is clinically useful in detecting hyperglycemia in women with prior HFDP at conventional diagnostic thresholds (hyperglycemia: sensitivity 84%; T2DM: sensitivity: 83%). The ability of a single HbA1c to differentiate between the two hyperglycemia classes, i.e., prediabetes and T2DM, was limited, with the test more accurate in predicting T2DM (specificity 97%)”.

was replaced with:

“Although not ideal, a single postpartum HbA1c at conventional diagnostic thresholds is clinically useful in detecting hyperglycemia in women with prior HFDP. The ability of a single HbA1c to differentiate between the two hyperglycemia classes, i.e., prediabetes and T2DM, was limited, with the test more accurate in predicting T2DM.”

- Some areas need to be clarified e.g. page 19 mentions that “women with prediabetes and HbA1c”

Replaced with “Women with OGTT prediabetes and HbA1c….” to clarify

>- 6.5% at higher risk for T2DM. Isn’t it DM with that HbA1c?

Once again, you make a valid point. If this had occurred in a clinical setting, it would have been diagnosed as diabetes mellitus (following a repeat HbA1c).

In this study, however, we did not classify primarily based on HbA1c, it was the comparator. Instead, women were primarily classified according to gold-standard OGTT. The HbA1c (comparator) was subsequently tested against the (classifier) OGTT. 

Nevertheless, "T2DM" was omitted from the revised sentence that now reads: “Women with OGTT prediabetes and a HbA1c ≥ 6.5% (48mmol/mol) represent a subset at higher risk (of T2DM) within the prediabetes range and may benefit from earlier pharmacological intervention with metformin”. 

Tables:

- P-values should be given for comparisons - added

Minor:

- Check consistency – authors using American English, yet haemoglobin spelt “haemoglobin” in

introduction. 

Addressed, thanks.

The following has also been addressed, thanks.

- NHLS does not need to be written out in full again under Biochemistry

- ADA in methodology needs to be written out

- HbA1c subheading in methodology – small c

- Don’t write out T2DM in full again in discussion

Reviewer 2 

Comment

A study from Tygerberg has shown that an HbA1c of >=6.1 may be more appropriate in the mixed ancestry population.

Did you look at this. HbA1c has several limitations in our populations such as the fact that it is affected by HIV , iron deficiency.

This should be mentioned

Response

The reviewer refers to research that we have been closely following and are familiar with. This manuscript already references the study the reviewer refers to in its discussion section (reference 50). A copy of the paragraph is provided below, as well as a yellow highlight of the applicable text. As requested by the reviewer, the phrase in italic was added as well. 

“Chivese and colleagues assessed the diagnostic accuracy of HbA1c vs FPG and OGTT in non-pregnant individuals from seven African countries.(49) The ethnic backgrounds resembled the ethnic representation in our cohort and included predominantly Black African women and women of Mixed Ancestry. In their study, the sensitivity of a HbA1c ≥ 6.5% (48mmol/mol) to diagnose T2DM was 58% vs OGTT, a value significantly lower than the 83% documented in our cohort. As expected, sensitivity improved by lowering the cut off [HbA1c ≥ 6% (42mmol/mol)] in their study, but this came at the expense of decreased specificity. Another recent study suggested that an HbA1c cut-off of 6.5% (48mmol/mol) may miss up to 50% of non-pregnant persons with T2DM, perhaps attributable to ethnic and population differences. (50) This study suggests an HbA1c ≥6.1% may be more appropriate in the mixed ancestry population.”

---

## [Editor Report · Decision Letter 1]

11 May 2023

Early postpartum HbA1c after hyperglycemia first detected in pregnancy - imperfect but not without value

PONE-D-23-03908R1

Dear Dr. Coetzee,

We’re pleased to inform you that your manuscript has been judged scientifically suitable for publication and will be formally accepted for publication once it meets all outstanding technical requirements.

Kind regards,

Jaya Anna George, BMBCH

Academic Editor

PLOS ONE
---

## [Editor Report · Acceptance letter]

1 Jun 2023

PONE-D-23-03908R1 

Early postpartum HbA1c after hyperglycemia first detected in pregnancy - imperfect but not without value 

Dear Dr. Coetzee:

I'm pleased to inform you that your manuscript has been deemed suitable for publication in PLOS ONE. Congratulations! Your manuscript is now with our production department. 

Kind regards, 

on behalf of

Dr. Jaya Anna George 

Academic Editor

PLOS ONE